# A Critical Review on Effect of Process Parameters on Mechanical and Microstructural Properties of Powder-Bed Fusion Additive Manufacturing of SS316L

**DOI:** 10.3390/ma14216527

**Published:** 2021-10-29

**Authors:** Meet Gor, Harsh Soni, Vishal Wankhede, Pankaj Sahlot, Krzysztof Grzelak, Ireneusz Szachgluchowicz, Janusz Kluczyński

**Affiliations:** 1Mechanical Engineering, School of Technology, Pandit Deendayal Energy University, Gandhinagar 382007, India; meet.gmtmm19@sot.pdpu.ac.in (M.G.); Harsh.smtmm19@sot.pdpu.ac.in (H.S.); Vishal.Wankhede@sot.pdpu.ac.in (V.W.); 2Faculty of Mechanical Engineering, Institute of Robots & Machine Design, Military University of Technology, 2 Gen. S. Kaliskiego St., 00-908 Warsaw 49, Poland; krzysztof.grzelak@wat.edu.pl (K.G.); ireneusz.szachogluchowicz@wat.edu.pl (I.S.); janusz.kluczynski@wat.edu.pl (J.K.)

**Keywords:** powder-bed fusion, process parameters, corrosion behavior, residual stresses, bio implant

## Abstract

Additive manufacturing (AM) is one of the recently studied research areas, due to its ability to eliminate different subtractive manufacturing limitations, such as difficultly in fabricating complex parts, material wastage, and numbers of sequential operations. Laser-powder bed fusion (L-PBF) AM for SS316L is known for complex part production due to layer-by-layer deposition and is extensively used in the aerospace, automobile, and medical sectors. The process parameter selection is crucial for deciding the overall quality of the SS316L build component with L-PBF AM. This review critically elaborates the effect of various input parameters, i.e., laser power, scanning speed, hatch spacing, and layer thickness, on various mechanical properties of AM SS316L, such as tensile strength, hardness, and the effect of porosity, along with the microstructure evolution. The effect of other AM parameters, such as the build orientation, pre-heating temperature, and particle size, on the build properties is also discussed. The scope of this review also concerns the challenges in practical applications of AM SS316L. Hence, the residual stress formation, their influence on the mechanical properties and corrosion behavior of the AM build part for bio implant application is also considered. This review involves a detailed comparison of properties achievable with different AM techniques and various post-processing techniques, such as heat treatment and grain refinement effects on properties. This review would help in selecting suitable process parameters for various human body implants and many different applications. This study would also help to better understand the effect of each process parameter of PBF-AM on the SS316L build part quality.

## 1. Introduction

Additive manufacturing (AM) offers plenty of advantages over the conventional manufacturing process, due to the layer-by-layer deposition of materials. The applications of additive manufacturing are increasing exponentially, especially in the medical [1] and aerospace industries [2], due to their unique feature of fabricating complex geometrical components. Powder-bed fusion AM uses thermal energy to selectively fuse powder particles on a powder bed as mentioned in the ASTM standard [3]. Laser powder-bed fusion (L-PBF) is one of the most studied AM processes for metal printing. The optimum process parameters are of utmost importance to obtain the desired properties. Many process parameters, such as powder characteristics, laser type, chamber environment, etc., are involved, and input process parameters make this process quite complex [4]. Hence, there is a need to optimize the process parameters to achieve desirable properties. Stainless-steel implants have been in use for years; however, with the conventional process, there are certain limitations regarding complex shapes and sizes. The implants made with AM techniques are free from conventional challenges and also provide better properties. Several researchers reported that with conventional processes, the metal ions dissolved in blood and urine after 10–13 bio implants in the human body [5]. Most of the implants failed, due to poor corrosion, fatigue, and wear resistance [6]. Hence, the improvement in the properties of SS metal implants is a promising area for researchers. Bio implants by additive manufacturing garner much attention because of their many advantages, compared to conventional manufacturing, such as ease of handling, complexity, minimal material wastage, and ease in operation [7]. There are always internal stresses induced in powder-based additive manufacturing, due to sequential heating and cooling [8]. The residual stress degrades the overall quality of the building part in terms of poor mechanical properties. Hence, the relation between the residual stress, input parameters, and properties need to be explored.

The entire study covers the effect of process parameters on mechanical properties, parameter effect on corrosion, and residual stress behavior. The literature collected over the past 10–15 years is on powder-based additive manufacturing of SS316L. The main four process parameters, i.e., laser power, scanning speed, hatch spacing, and layer thickness, and their influences on mechanical properties are considered. In the first section, the effect of the process parameters on tensile strength, hardness, and porosity with microstructure evolution is explained. In the second section, the corrosion and residual stress behavior of additively manufactured SS316L with process parameters are described. To conclude, all the reported process parameters on AM-SS316L and their influences are considered. This study will help to define the process parameter window for future applications to obtain desired strength and hardness. This study also helps by collecting all the process parameters effects reported till date for LPBF AM of -SS316L, this collected data can be used for future work in machine learning for the optimization of process parameters.

## 2. Process Parameter Selection

### 2.1. Tensile Strength

Tensile strength is one of the important factors considered to evaluate the quality of the build part. In AM, many parameters are involved at a time. Hence, it is essential to observe the effect of each parameter on tensile strength. This section covers powder selection, all the AM input parameters, the optimization of parameters, chamber environment, post-processing techniques, and comparison of tensile strength with other manufacturing processes in terms of properties and microstructural point of view. The selection of powder size plays a crucial role in the output mechanical properties. The smaller powder particles produce a denser product with minimal porosity and decreased chance of internal defects. Chen et al. [9] examined the effect of powder particles on mechanical properties with an energy density of 55.55 J/mm^3^ and observed that a finer particle of around ~16 µm gives the highest tensile strength in the range of 610 MPa as shown in Figure 1. The fracture surface with dimples as shown in Figure 1(A_2_,B_2_) also implies the ductile failure of the sample.

The layer-by-layer deposition gives sequential heating and cooling, which results in a finer austenitic grain structure, compared to other conventional manufacturing processes. Hajnys et al. [10] investigated the optimization approach, using Taguchi to establish the relation of process parameters on mechanical properties. They stated that the most influencing parameter for tensile strength of AM build parts is the scanning speed followed by the scanning pattern and laser power. They obtained maximum and minimum tensile strengths at 650 mm/s and 1200 mm/s scanning speeds, respectively. Wang et al. [11] also reported that at lower energy density, due to insufficient melting, the viscosity of the liquid melt pool is poor. Hence, the spreading of liquid metal is difficult, which results in porosity. At a higher energy density level, the liquid formation is enough for uniform spreading of liquid metal, which makes a denser structure. They observed that at energy density of 125 J/mm^3^, the highest tensile strength around 590 MPa was achieved. The fracture morphology at different energy density is shown in Figure 2. The chances of large crater voids are high, and cracks propagate easily, due to local brittleness at lower energy levels. The fracture morphology is different at different energy density inputs. At the energy density of 125 J/mm^3^, more densification can be observed in Figure 2B. Hence, the tensile strength increases, and the further increase in energy density makes larger and shallower dimples, which reduce the tensile strength.

The data of the effect of different process parameters on the as-built tensile strength of SS316L are collected from the literature as shown in Table 1. The observation from collated data is plotted concerning different parameters. The overall influence of energy density on ultimate tensile strength is shown in Figure 3. The graph shows that the tensile strength increases with incremental increase in the energy density, due to better densification. However, at a higher energy density, keyhole porosity leads to restrict the further increment in tensile strength. The optimal energy density for the better tensile property is in the range of 50 J/mm^3^ to 105 J/mm^3^.

The microstructure evaluation has a vital role in the mechanical properties in particular; the strength of the component depends on grain size and dislocation density. The refined grain structure greatly influences the porosity, residual stress, and strength of the components. The SLM process generates a finer microstructure, compared to the direct energy deposition (DED) process, due to more grain refinement. The SLM microstructure contains a higher dislocation density, compared to DED and other conventional processes. The dislocation hinders the movement of discontinuity. The heat treatment reduces the strength from 671 MPa to 616 MPa in the SLM part and 645 MPa to 600 MPa in the DED build part [12]. The heat treatment does not show much improvement in the mechanical property of SS316L, as phase transformation does not occur. The best alternative to heat treatment is the solid-state grain refinement process, which improves the strength along with ductility [13]. The preheating of the base plate up to a certain temperature gives a lower thermal gradient, which provides good mechanical property and denser part with minimal defects. Zhang et al. [14] investigated the effect of preheating temperature and build orientation on density, strength, and deformation. The tensile strength specimen for vertical build-up and horizontal build-up is shown in Figure 4. They obtained the highest strength and dense part at the 150 °C preheating temperature.

The build orientation and location in the build plate also affect the strength of the component, as the fracture behavior is different when horizontal vs. vertical [18,19]. Casati et al. [20] plotted the stress–strain behavior of tensile specimens built vertically and horizontally with 580 MPa and 684 MPa of UTS. This can be attributed due to loading different loading directions in vertical and horizontal build parts. The higher scanning speed creates a small melt pool with lower wetting characteristics, leading to separate solidification, popularly known as the ‘balling effect’, and a highly porous structure. The highest tensile strength (650 MPa) could be obtained with a scanning speed of 90 mm/s, whereas a higher scanning speed of 180 mm/s gave poor mechanical properties [21]. Li et al. [22] used the same approach to produce a component with gradual property increment, which is mostly used in biomedical applications. The layer thickness plays a crucial role in the strength and surface finish of the build part. Delgado et al. [16] examined the effect of parameters on mechanical properties for DMLS and SLM. They observed that as the layer thickness decreases, the partial re-melting of previous layers occurs, making proper diffusion between the layers. Hence, the maximum strength 580 MPa is reported with 30 µm of layer thickness. Reddy et al. [23] approached the small-scale testing sample for AM, as the overall cost of DMLS is higher, and compared the characterization with the standard sample size. They reported that the tensile strength of SS316L is 500 MPa and 516 MPa for small scale and standard size samples, respectively.

The effect of various process parameters on the tensile strength of the building part is discussed in the above section. The overall most influencing factor is energy density for SS316L, and the highest tensile strength is found at in the energy density range of 50 J/mm^3^ to 105 J/mm^3^. The other individual parameters that play an important role are the scanning speed and layer thickness. The maximum tensile strength of SS316L obtained is 712 MPa with powder-based AM. However, commonly, the tensile strength varies from 600 MPa to 650 MPa. A significant improvement in the tensile strength can be obtained with grain refinement processes rather than heat treatment.

### 2.2. Hardness

Hardness is the most important surface property considered in many industrial applications. In this section, each process parameter’s influence on hardness is discussed. The influences of the powder particles, laser power, energy density, build direction, scanning pattern, hatch spacing, porosity, and build chamber environment as well as the effect of various post-processing techniques are covered.

The particle size alone does not show much effect on hardness; rather, it more depends on the scanning pattern and varies in each build direction. Chen et al. [9] found that for finer particles, the hardness in the XY cross-section (276 HV) is lower than the hardness in the YZ (286 HV) and XZ (291 HV) cross-sections. Pannitz et al. [24] performed a comprehensive study on five different powders available of SS316L at standard process parameters and found that the achieved average hardness is 195.4 HV. The hardness value is strongly correlated with the porosity. Hence, the parameters which reduce the porosity will result in higher hardness. Tucho et al. [25] investigated the effect of porosity, scanning speed, and hatch spacing on hardness. They reported that as the energy density increases, the hardness of the component increases linearly up to a certain level and decreases due to gas trapping pores generated at a higher energy density. The maximum and minimum hardness were observed as 213 HV and 176 HV for the energy density of 80 J/mm^3^ and 50 J/mm^3^, respectively. The hatch spacing also affects the hardness level; the minimum hatch spacing of 80 µm gives the maximum hardness for the 80 J/mm^3^ energy density. The data of the effect of different process parameters on as build hardness of SS316L are collected from the literature as shown in Table 2. The observation from the collated data is plotted concerning different parameters, such as hatch spacing, layer thickness, and energy density. The overall effect of hatch spacing on hardness from the collected literature data is shown in Figure 5 [24]. The lower hatch spacing contains more overlap in between laser scan tracks that helps to achieve complete fusion between tracks, whereas higher hatch spacing involves more porosity defects. The better hardness value is achieved at hatch spacing of 80 µm.

The layer thickness also influences the hardness value in the building part and at the lower layer thickness level; the bond between each layer is stronger, which leads to higher hardness. The maximum hardness reported at the layer thickness of 30 µm is shown in Figure 6. The lower layer thickness promotes refinement of the underlying deposited structure, which improves the hardness. On other side, lower thickness also consumes more time to deposit the same structure.

Wang et al. [11] also reported that with an increase in the energy density, the hardness increases up to a certain level after which it reduces due to coarsening of the grains. They examined that the maximum micro hardness of 281.6 HV is obtained at 125 J/mm^3^ and further increasing the energy density leads to a reduction in hardness. The overall effect of energy density on hardness from the collected literature data is shown in Figure 7. The increment in hardness with respect to higher energy density is related with the better fusion between and layers and tracks. Further reduction in hardness at a higher energy density leads to keyhole porosity.

Tolosa et al. [29] also reported that as the energy density increases, the hardness increases up to 225 HV at 125 J/mm^3^**.** The hardness highly depends on the scanning speed, wherein the higher scanning speed leads to a faster cooling rate, which results in a finer grain and increases the hardness. Sun et al. [28] experimented with the productivity of the SLM process at a higher scanning speed. The hardness value was found to be in the range of 213 HV to 220 HV, which is greater than that of the SS316L annealed part. The reason for the higher hardness is due to nano-amorphous inclusions (known as silicate), which increase the dislocation density. Inclusion depends on the oxygen contained in the build chamber, which generally should not be more than 0.05% [28]. Cherry et al. [26] also reported that the optimum values of laser power and energy density could increase the hardness value. Delgado et al. [16] investigated the effect of process parameters on hardness and reported that the hardness value decreases as the build direction changes from 0’ to 90’. The increment in the layer thickness and scanning speed also reduces the hardness. Hitzlerr et al. [17] reported that hardness does not show much variation for in-plane anisotropy for the different scanning strategy. Yusuf et al. [15] stated the vital result that the hardness value does not change with different build directions. However, hardness is different for the different scanning directions. They measured the value of hardness for the XY, YZ, and XZ planes and found 262 HV, 239 HV, and 237 HV, respectively. Revilla et al. [30] also showed a similar nature for the build orientation anisotropy of hardness. The hardness in the perpendicular direction is 289 HV, which is higher than the average value of hardness in the parallel direction, i.e., 272 HV. Saeidi and K et al. [31] also studied that in the AM process, due to finer grain size, the dislocation density of the austenite cell is higher; hence, the strength and hardness could be achieved to be higher than the conventional processes. however, porosity must be optimized in a controlled manner. Muley et al. [13] also reported that as the grain size decreases, hardness increases. The warm multi-axially forged process increases the hardness of AM build parts, due to the ultra-fine grain structure. Yusuf et al. [15,32] investigated the effect of grain refinement on mechanical properties of SS316L. They used a high-pressure torsion technique on an AM build part and achieved almost twice the hardness, from 250 HV to 525 HV, due to refinement of the cellular structure, which effectively reduces porosity. Pagáč et al. [33] investigated the effect of process parameters on hardness and concluded that laser power and heat treatment do not influence the printed parts. The hardness of the printed part was found to be around 50 HRB higher than the rolled part. However, the heat treatment tends to reduce the hardness, due to recrystallization. Chimmat et al. [34] compared the effect of heat treatment at 650 °C; expectedly, hardness reduces to 185 HV from 210 HV. Reddy et al. [23] reported a similar trend that after the heat treatment, due to stress release, the hardness value drops by 12%. The tempering temperature also contributes to determine the hardness of the component, as the percentage of retained austenite influences the hardness value. Chen et al. [35] experimented with the SS316L at different tempering temperatures and found that the maximum hardness of 620 HV could be achieved between 427 °C and 500 °C tempering temperature.

In this section, the influence of the process parameters on hardness is considered. The overall hardness with the powder-based AM of SS316L is achieved in the range of 220 HV to 270 HV depending upon the process parameters. The most affecting parameters contributing to hardness are hatch spacing, scanning pattern and layer thickness. The overall energy density range for optimum hardness is 80 J/mm^3^ to 125 J/mm^3^. Hardness is the anisotropic property and varies in each different build plane direction. Heat treatment reduces the hardness value, due to recrystallization. The various surface modification techniques, which refine the grain structure, improve the hardness significantly.

### 2.3. Porosity

The porosity of the built part is one of the important properties to be considered, as, directly or indirectly, it influences all the mechanical and microstructure behaviors. In additive manufacturing, densification of the material is the most crucial factor to be considered. The optimum range of process parameters is required to obtain a denser product. In this section, the formation of pores, affecting process parameters and their influence, optimizes parameters in terms of mechanical and microstructural properties are discussed.

The porosity is generally classified into two categories, namely, metallurgical pores and keyhole pores. The metallurgical pores are generated, due to the oxygen content and other inclusions, in 100 µm with a spherical shape. The keyhole pores occur at a higher scanning speed. Generally, keyhole pores formed irregular shapes of more than 100 µm in size as shown in Figure 8 Metallurgical pores are not very dependent on process parameters, whereas keyhole pores can be controlled by controlling the energy density. Pagáč et al. [33] explained that a process with high laser power could give lower porosity at a constant scanning speed. On the other hand, the annealing heat treatment tends to expand the pore and thus, increases porosity.

The finer powder particles generate a denser product, as more surface area is available to absorb the laser energy, resulting in the complete melting of particles. The highest density (99.99%) is reported with a particle size range of 5 to 40 µm [9]. The porosity and distribution of pores is the main factor that affects the mechanical property of the build part, rather than the influence of energy density in SLM of SS316L. Wang et al. [36] demonstrated that, even with constant energy density, a porosity above 0.115 mm results in a drastic reduction in the mechanical properties, due to an increase in the stress concentration around the pores, which reduces the ductility.

The optimum energy density is the critical parameter for the dense structure and is achieved by the essential selection of the laser power, scanning speed, hatch spacing, and layer thickness. The part produced with a lower energy density may have unmelted particles, which results in a porous structure. The higher energy density creates a steep “V” shaped melt pool, which leads to micro evaporation from the melt pool; that gas is trapped below the melt pool, which again results in porosity. Tucho et al. [25] experimentally obtained the energy density levels for minimum and maximum porosity for SS316L as 80 J/mm^3^ and 50 J/mm^3^, respectively. The SEM images of porosity shown in Figure 9 also conclude that the scanning speed directly affects the porosity level at lower energy density levels; as scanning speed increases, the porosity increases but at higher energy density, the scanning speed does not affect the porosity. The minimum porosity up to <0.03% was observed at 75 J/mm^3^ energy density.

Cherry et al. [26] also investigated the importance of optimum laser energy density for minimum porosity. They observed a similar trend; as the energy density increases, porosity increases up to a certain level after which it decreases. The minimum and maximum were obtained at 104.52 J/mm^3^ and 41.81 J/mm^3^, respectively. The further increment in the energy density to 209.03 J/mm^3^ increases the porosity, due to the evaporation of microelements as shown in Figure 10 [28]. The data of the effect of different process parameters on the relative density of SS316L are collected from literature as shown in Table 3. The overall effect of the energy density on the build part density from the collected literature is shown in Figure 11, where the process window for the better density of SS316L includes using an energy density of 80 to 105 J/mm^3^.

Li [22] also observed that a theoretical density (96%) could be obtained with 90 mm/s scanning speed, whereas if the scanning speed increases to 180 mm/s, density drastically reduces to 65%. The porosity of the build part strongly depends upon the melt pool size variation, which is correlated with the laser power and scanning speed directly. Li et al. [22] also correlated the melt pool variation with the porosity. The higher scanning speed creates a very small melt pool, leading to a higher porosity with a lower tensile strength structure, as shown in Figure 12. This phenomenon could help build the gradient properties component, such as bio implants, where the scanning speed variation produces the gradient porosity component.

They reported that the most influential parameters on the density of AM build parts are the scanning speed, scanning pattern, and laser power [10]. They validated that the maximum and minimum densities obtained are 99.96% and 75.52% at 650 mm/s and 1200 mm/s, respectively. Kruth et al. [31] reported the effect of the scanning speed on porosity. The results conclude that at a lower scanning speed, the grain size increases and the melt pool forms an irregular shape, resulting in the large size of the pores. The overall impact of the scanning speed on the porosity of the build part from the collected literature is plotted below in Figure 13.

The porosity could also be due to the gas atomization of the powder material. Yusuf et al. [15] investigated the X-ray computed tomography (XCT) analysis for analyzing pore distribution. They observed that the scanning pattern also influences the pore size and distribution. The pores are located at the islands’ interface region. The average length of the pores also in the expected zone of 5 µm to 45 µm for 60% of the pores concludes that in SLM of SS316L, a highly dense structure (>99%) could be achieved. The vertical cracks contribute significantly to lower density and poor mechanical properties; thus, the scanning strategy must be selected accordingly, increasing the power input [36,37]. Dewidar et al. [38] also experimented with the porous structure for bone implants with 40% to 50% porosity, which has the desire elastic modulus of 26 GPa for the human body. Although the higher level porosity contributes to crack growth and failure mechanisms, the micro-pores’ uneven distribution also plays a crucial role in the corrosion behavior [30]. Porosity also affects the creep life of the component. Dao [39] explained that the porosity exaggerates the cavitation effect, leading to reduced creep resistance. AlFaify [40] performed the optimization of the process parameters with the regression model and found that layer thickness and point distance are the parameters that significantly affect the build part’s porosity, whereas hatch spacing is the least affective for porosity.

As discussed in this section, porosity is observed at higher and lower energy density levels. The optimum range for denser products is achieved in the range of 80 J/mm^3^ to 105 J/mm^3^. Each process parameter’s effect on a denser product is discussed; the scanning speed, laser power, and layer thickness are important parameters to be considered. The denser product above 99% density is observed at a 650 to 800 mm/sec scanning speed depending upon the energy density level. The general pore structure is still observed in powder-based AM in the range of 5 to 40 µm, even at a 99% dense structure.

## 3. Corrosion Behavior of SS316L in the Human Body

In this section, the corrosion behavior of SS316L bio implants in additive manufactured parts is discussed. Musculoskeletal injuries and musculoskeletal diseases are increasing day by day in the human body. The estimated general surgeries of orthopedic implants are 1 million per year and are estimated to increase by 2 million per year in 2025 [41].

Metal implants are used in different body parts, including as bone replacements, knee joint replacements, dental implants, screws, plates, etc., as shown in Figure 14 [43]. Metal implants in the human body are likely to degrade in the body environment. Human body fluid contains various ions and organic particles, which readily react with the fitted implants. The pH value of the body is generally 7. However, it varies many times, due to acidity or any diseases or infections. This acidic environment also can be the reason for deviation in implant properties [44]. Various biodegradable metals are used in bio-implant applications, such as austenitic stainless steel, titanium-based alloys, cobalt-based alloys, and magnesium-based alloys. SS316L and Ti-6Al-4V are the most common materials used in different bio implant applications [45,46]. Metal implants have been used for years now, but still, there are specific challenges that were reported in recent years. Okazaki and Gotoh [47] reported that after 10–13 years, hip joint implants in the human body degraded and metallic ions were found in the human blood and urine. Most of the implants fail due to fracture, corrosion, and wear in the human fluid environment. Sivakumar et al. [48] reported that 90% of SS316L implants failed, due to corrosion attacks in human body fluid. Two characteristics explain the corrosion phenomenon for steel implants: the first is the oxidation and reduction process with the environment, and the second is oxide layer formation, which prevents corrosion. Whenever these oxide layers are deformed, the surface starts to react with the environment and eventually degrades. This behavior is called passivity, and the oxide layers are called a passive layer [49]. The stability of the oxide layer in human body fluid is the most essential to prevent corrosion. Several types of corrosion failure were observed for bio implants: pitting corrosion, fatigue corrosion, stress corrosion, and inter-granular corrosion [49]. Corrosion is the most affecting factor in implant failure; however, wear and fretting, surface cracks, and porosity also contribute to the failure of a bio implant. The combination of all mechanisms leads to early failure of the implant. Hence, the surface properties and mechanical properties both are equally important. The new layer-by-layer additive manufacturing techniques have started developing customized and complex implants with ease [50]. The AM build part of SS316L has different mechanical and microstructural properties, compared to the conventional manufacturing process. Hence, many researchers have started to explore the corrosion behavior of SS316L implants fabricated using AM. The corrosion performance measured with a similar environment to human body fluid to reflect the same situation is called in vitro analysis. The various fluids developed in laboratory to simulate the same behavior are called simulated body fluids, such as Ringer’s solution, Hank’s solution, phosphate buffer saline (PBS), human serum and other physiological simulated body fluids [50]. The general corrosion behavior of a material is evaluated with a standard testing procedure (according to ASTM F2129 or ASTM G 61) called potentiodynamic polarization [49,51,52,53,54]. Many researchers compared the corrosion behavior of SS316L with AM and wrought alloy in simulated body fluids. Revilla et al. [30] reported interesting work by comparing the corrosion behavior of SS316L builds by different AM processes. Their work concluded that the highest corrosion resistance was found in the SLM part followed by LMD and wrought alloy as shown in the polarization curves of Figure 15. This phenomenon can be attributed to different microstructure evolution with SLM and DED, where the much finer structure in SLM shows better corrosion performance. Durejko et al. [55] investigated the corrosion behavior of an SS316L build by a laser net shaping DED process and found lower pitting resistance compared to conventional manufacturing because of chromium segregation and the dual-phase microstructure.

Lodhi et al. [56] compared the corrosion behavior of SLM and wrought alloy in three different simulated environments of human serum, PBS, and 0.9 M NaCl. In all the other conditions, the AM part had the highest breakdown potential as shown in Figure 16. These can be attributed to the refined microstructure in AM, which has better passive layer stability [56].

The same group reported another study of corrosion behavior of AM SS316L in an acidic environment for less than 3 pH values. The microstructure of wrought alloy vs. AM is shown in Figure 17. It is observed that the heterogeneous microstructure for wrought alloy and very finely distributed sub-granular structure exists in the AM part [56,57,58]. Suryawanshi et al. [59] also reported a similar study, comparing the corrosion behavior of SLM and wrought alloy for different materials. Due to grain refinement in the SLM part, better corrosion resistance was observed.

Al-Mamun et al. [60] also compared the corrosion resistance of wrought SS316L, and AM builds part in the physiological environment. The AM build part shows better passive film stability, which restricts the pit formation to a higher potential of −67 mVSCE, where wrought SS316L has −204.3 mVSCE. The AM part has a refined microstructure with a uniform distribution of Cr. The absence of Mns inclusion leads to fewer and smaller pit formations than wrought alloy as shown in Figure 18.

Another study reported that AM part performs well in a corrosion environment, compared to wrought alloy, due to a layer-wise refined microstructure, the absence of MnS inclusions, and more robust passivity behavior [61,62]. Nie et al. [62] also reported the analysis of film stability in the AM part in the environment of a 3.5 wt% NaCl solution. The dual-phase passive film containing oxides and hydroxides of Cr and Fe was observed in the AM part and wrought alloy. High corrosion resistance was found in the AM part because of the thicker and more stable passive film found in the AM part. The effect of the grain size on corrosion behavior showed some contradictory results. Ralston et al. [63] reported that sometimes, a finer grain structure increases the corrosion behavior because as the boundary density increases, better adhesion of the passive film is observed. Sometimes the passive layer formation depends upon microelement distribution and environmental conditions. Hence, with AM implants, some contradictory results were also reported. The AM implants show better corrosion resistance, compared to the conventional manufacturing process [64]. This can be attributed to the unique heating and cooling mechanism in layer-by-layer manufacturing. The higher cooling rate of around 107 K/sec leads to a finer microstructure, rapid boundary diffusion, and uniform element distribution. Hence, less time for Mns nucleation reduces the Cr-depleted zone in AM of SS316L [63,64]. The various post-processing techniques were also examined to improve the corrosion resistance and surface property of bio implants of SS316L [65,66]. Zhou et al. [67] used the post-annealing of an SS316L build part to enhance corrosion resistance. They found that intergranular corrosion starts from the melt pool boundaries in an as-built sample as shown in Figure 19. As melt pools are dissolved in the heat-treated sample due to recrystallization, corrosion initiates at the multiple grain boundaries. The heat treatment at recrystallization ultimately reduces the corrosion resistance [68].

A similar study by Laleh et al. [69] also reported that high-temperature post-processing at 1100 °C and 1200 °C increases the tendency of pitting corrosion at multiple sites. Benarji et al. [68] compared the corrosion resistance at different solution treatment temperatures, 1073 °C and 1273 °C, of SS316L. Figure 20 shows the SEM images of both conditions and it can be observed that the ferrite phase dissolved at a higher temperature. In the only austenitic microstructure, the corrosion rate is higher, compared to a Cr-rich ferrite microstructure.

The overall scenario of an additively manufactured bio implant of SS316L is discussed. Corrosion is a major concern for the early failure of a bio implant. The manufacturing defects, critical environment, microstructure evolution, surface, and mechanical properties play an important role in the life of a bio implant. Additive manufacturing implants have a positive impact on corrosion behavior, as discussed. However, more research is required on microstructural evolution, the effect of different phases in the microstructure, parameter optimization for corrosion properties, and residual stress influences on corrosion behavior.

## 4. Residual Stress

In this section, the residual stress formation phenomenon in powder-based additive manufacturing explains their types and measurement techniques. The effect of process parameters, such as energy density, laser power, layer thickness, and build part orientation on the formation of residual stresses are elaborated. The various post-processing techniques and optimum parameters to reduce the amount of residual stress are explained.

The main consequence of additive manufacturing is internal residual stress generation in the process. The layer-by-layer continuous heating and cooling cycle leads to generate thermal stress in the build component. This residual stress reduces strength, surface cracks, delimitation from the base plate, and distortion, and results in decreased overall quality of the part [68]. There are three types of residual stress generated in the build part, namely, Type-1, Type-2, and Type-3. The stress generated at the macro level inside the build part due to thermal deformation is termed Type-1. The stresses generated at the micro level due to grain orientation and anisotropy at the atomic scale are called Type-2 and Type-3, respectively [69]. The residual stress generation phenomenon is dependent upon two cycles of heating and cooling. There are three different zones developed in the heating and cooling phase: the melting zone, heat-affected zone, and unaffected zone. In the heating phase, the powder exposed by a laser beam is heated to a higher temperature and tries to expand. Since the surrounding powders are colder, they try to restrict expansion. Hence, where the laser beam is spotted on those locations, compressive residual stress is generated [70,71]. Once the laser beam is passed from a particular location, the second phase begins a cooling phase. In the cooling phase, the high-temperature zone cools down immediately and shrinks. The previous plastic strain and bonding again restrict this shrinkage of the material with surrounding particles. Hence, in the cooling phase, tensile residual stress is generated. This tensile and compressive residual stress forms in every new layer deposition. This phenomenon of continuous heating and cooling is explained in Figure 21 [72]. The overall mechanism occurs is as more numbers of the layers are deposited, the lower layers try to expand, and the top layers try to compress [73]. Another mechanism occurs when the first layer is deposited on the cold surrounding deposited material, which shrinks immediately but contraction is restricted by the colder base plate and surrounding material. Hence, tensile residuals are generated on the initial layers, whereas the base plate compressive residual stress is generated [74].

The maximum residual stress in the yield strength range generally occurs at a higher temperature gradient, i.e., joint of the base plate and first layer [75]. When the build part is removed from the base plate, residual stresses are reduced drastically due to minor deformation [76,77]. Hence, the overall trend of residual stress after removing the base plate is found to be similar by many researchers, as follows. The lower portion of the build part contains tensile residual stress, and further, as height ingresses, it converts into the compressive residual zone. Then, again, the top layers contain tensile residual stresses [73]. The value of residual stress also varies with the height of the build part; as more layers are deposited, heat accumulates in the deposited layers. The temperature gradient reduces with the height of the build part, leading to a reduction in compressive residual stresses. The residual stresses increase with the energy density, as significant molten pool results in more volumetric shrinkage. The density of the build part also affects residual stresses. Residual stresses are found to be lower with a more porous build part [38]. Manojakumar et al. [34] reported that optimized process parameters for density produce residual stress that is equal to half of the yield strength. They also reported that post-heat treatment of SS316L revealed residual stress compared to that in Figure 22 [14].

The lower laser power process involves a higher cooling rate, which eventually increases the temperature gradient and residual stresses [78]. Tong et al. [79] reported that the laser power has a great influence on internal stress generation. As the laser power increases, residual stress first decreases and then increases. This phenomenon can be attributed to the cooling rate and melt pool formation. The increment in laser power reduces the amount of residual formation to some extent. The higher laser power also gives more extensive melt pool formation, which leads to more volumetric shrinkage. Yang et al. [80] explained the effect of the build height, direction, scanning pattern, support material on residual stress. They reported that the lower heat input and small line length produce low residual stresses. Mukherjee et al. [81] explained the effect of layer thickness and heat input on residual stress formation. The lower layer thickness reduces residual stresses by 30%. The lower thickness involves higher exposure time, and it reduces the overall temperature gradient. The highest residual stresses were generated at the end of the tracks. This is why the build part is to be warped or delaminated from the base plate [82,83]. There are several destructive and non-destructive methods used to measure the residual stress, such as the hole drilling method, X-ray diffraction, neutron diffraction, crack compliance method, and magnetic Raman [84,85,86,87].

As discussed in this section, residual stress cannot be eliminated in layer-by-layer manufacturing. Several techniques can be used to reduce the residual stresses, such as preheating of the base plate, re-scanning of the tracks [85], post-heat treatment of the build part [86], laser shot pinning, the addition of dwell time between layer deposition, optimized orientation, and the location of the build part [74,75,87,88,89,90,91,92]. Shiom et al. [93] reported that re-scanning effectively reduces residual stress by 55%, and preheating the base plate can reduce residual stresses by about 40%. Further, residual stress on other surface characteristics, such as corrosion behavior [49], wear rate, and microstructure evolution reported some controversy. Additionally, there are some new types of approaches in the measurement of residual stress of AM part suggested by Kluczyński et al. [94,95], which significantly affect the fatigue properties of AM-made 316L steel [96,97,98], which was shown in the co-authors’ own work. Hence, more work is required to explore the residual stress influences on corrosion behavior, wear rate, and microstructure evolution.

## 5. Conclusions

The powder-bed fusion additive manufacturing of SS316L for different process parameters and their influences on different mechanical and surface properties were extensively covered and critically discussed in this review. The effect of main input parameters of powder-bed fusion AM, such as laser power, layer thickness, hatch spacing, and scanning speed on build part properties is covered. The effect of post-heat treatment, grain refinement techniques on build properties were also discussed. Each parameter influences tensile strength, hardness, and relative density of the build part and are individually critically discussed. The microstructure evolution for AM-built parts and their properties comparison with different AM processes are also studied. This study helps to identify the process parameters window to obtain desired properties. The key concluding points from the entire literature are as follows:The tensile strength of PBF-AM of SS316L is found to be in the range of 600 MPa to 650 MPa. The maximum tensile strength reported for SLM builds part is 712 MPa. The energy density input is required in a range of 50 J/mm^3^ to 105 J/mm^3^ to obtain significant tensile strength. The most influencing parameters affecting the tensile strength of the building part are scanning speed and layer thickness. The tensile strength can be improved with various post-processing techniques, such as grain refinement. The pre-heating of the powder bed also improves the tensile strength of the as-built sample.The hardness value of the as-built sample of the powder-bed AM of SS316L is observed in the range of 220 HV to 270 HV. Hardness is an anisotropic property, observed to be different in the different build directions. The energy density input for the better hardness of the as-built sample is observed in a range of 80 J/mm^3^ to 125 J/mm^3^. The most influencing input parameters for hardness are hatch spacing and the scanning pattern. The various grain refinement techniques can improve the hardness of an as-built sample.In layer-by-layer manufacturing, higher relative density or lower porosity is the first objective to improve the overall strength of the building part. All the different mechanical properties and electrochemical behaviors are influenced by the amount and distribution of micro pores in the building part. The optimum range of energy density input in powder-bed AM can produce a denser structure. The optimum energy density window to obtain the maximum denser product is in the range of 80 J/mm^3^ to 105 J/mm^3^. The most influencing parameters to obtain a denser product are scanning speed, laser power and layer thickness.The different electrochemical behaviors of the PBF-AM build SS16L part are discussed. The corrosion behavior is observed to be different, due to layer-wise microstructure evolution in AM. The process parameters, environment, micro elemental segregation in AM, and different passivation behaviors play an important role to decide the life of a bio implant in the human body. The AM shows better corrosive resistance in the human body; however, the influence of each process parameter, microstructure evolution, the effect of different phases, and elements studied in this direction are required to better understand corrosion behavior of AM parts.Residual stress is the most affecting factor to degrade mechanical properties of an as-built part of AM. The formation of residual stresses and their influences was covered. However, further studies are required on the effect of residual stresses on corrosion behavior, wear rate, and different surface and structural properties for better understanding the correlation with other input process parameters.

## Figures and Tables

**Figure 1 materials-14-06527-f001:**
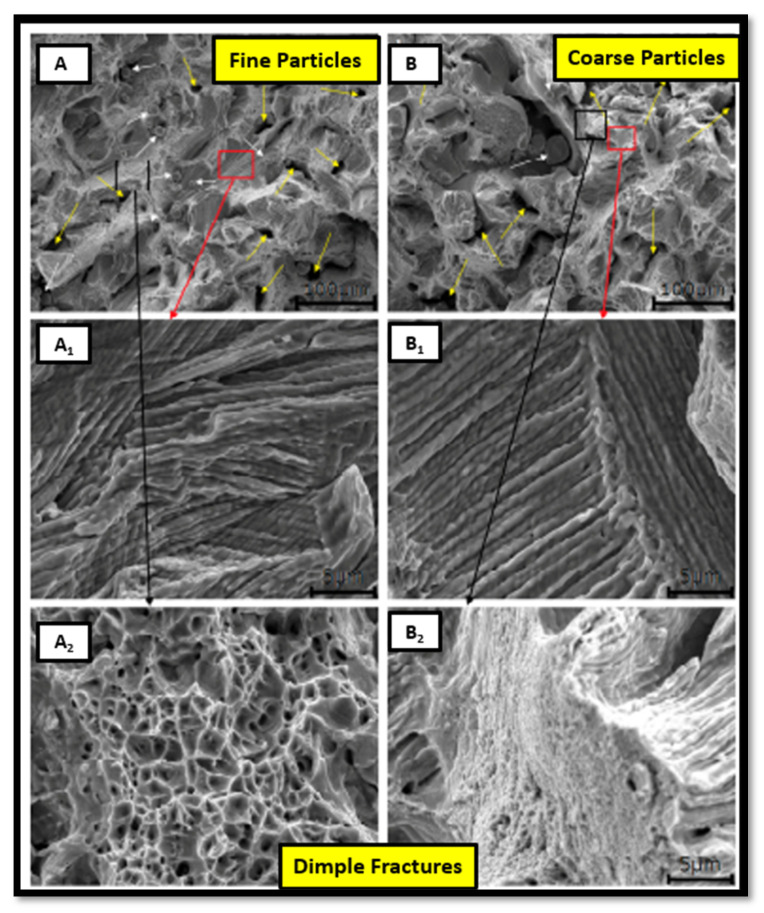
Fracture surfaces. (**A**) Finer and (**B**) coarse powder particle, (**A_1_**,**B_1_**) sub-grain micrographs for finer and coarse particles, (**A_2_**,**B_2_**) fracture surfaces for finer and coarse particles with dimples [1].

**Figure 2 materials-14-06527-f002:**
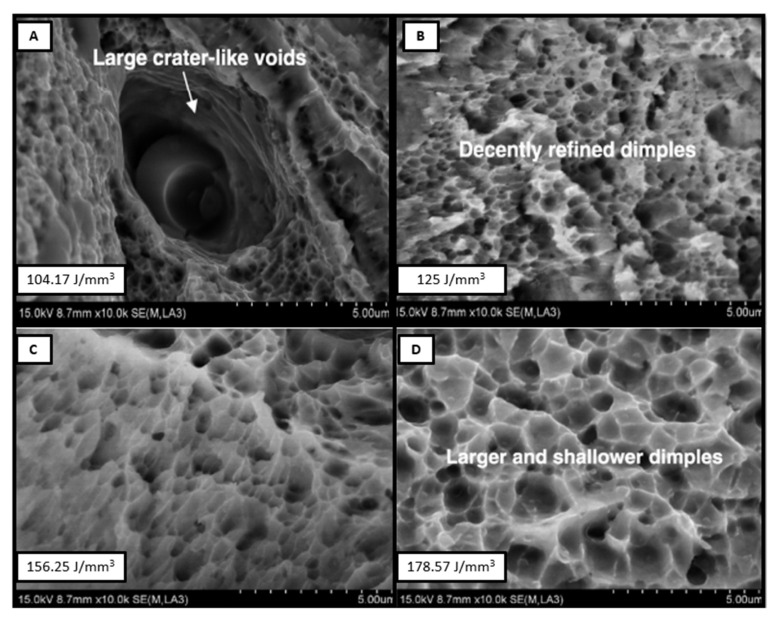
Fracture morphology at different energy densities (**A**) 104.17, (**B**) 125, (**C**) 156.25 (**D**) 178.57 J/mm^3^ [4].

**Figure 3 materials-14-06527-f003:**
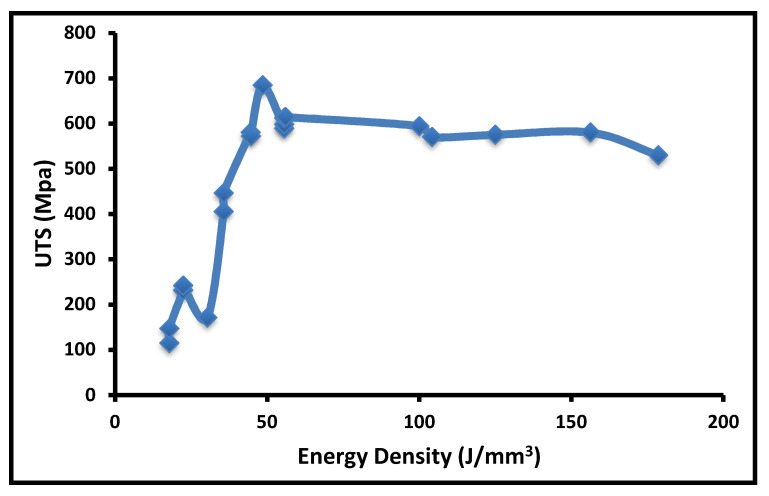
Tensile strength at a different energy density.

**Figure 4 materials-14-06527-f004:**
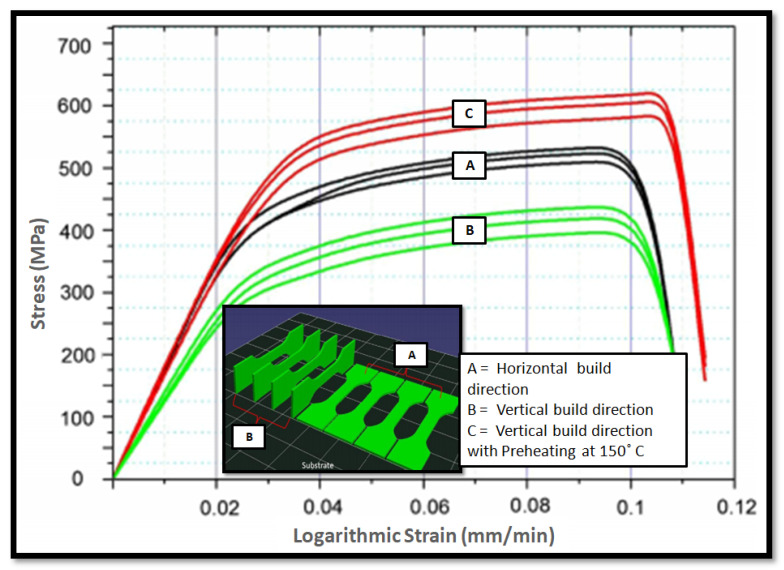
Tensile specimen build HZ and vertical direction with respective stress–strain curve [7].

**Figure 5 materials-14-06527-f005:**
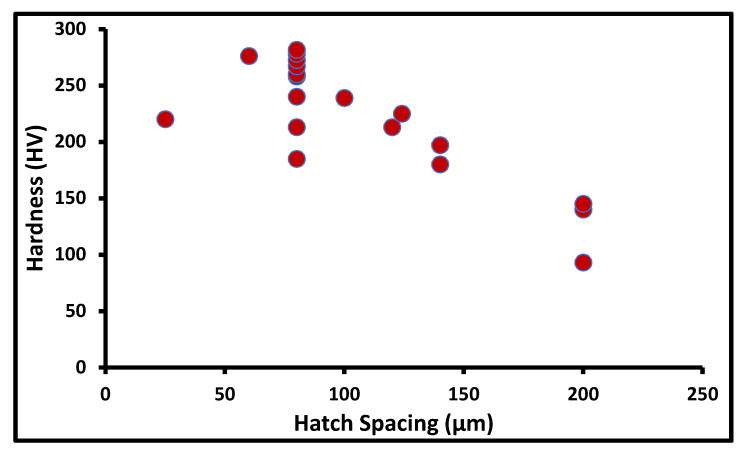
Hardness at varied hatch spacing.

**Figure 6 materials-14-06527-f006:**
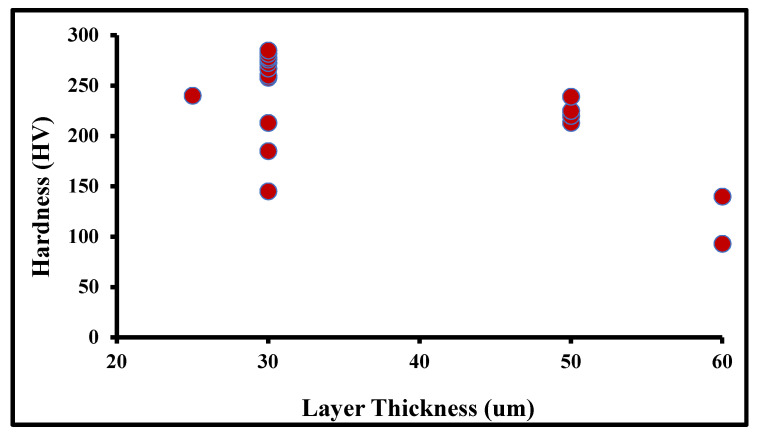
Hardness at varied layer thickness.

**Figure 7 materials-14-06527-f007:**
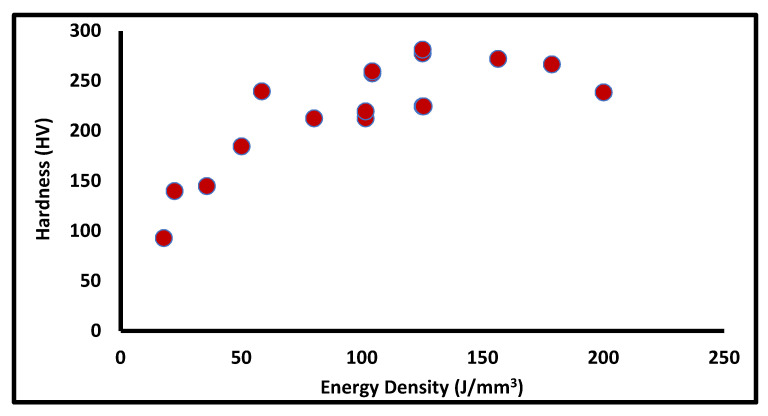
Hardness at different energy density.

**Figure 8 materials-14-06527-f008:**
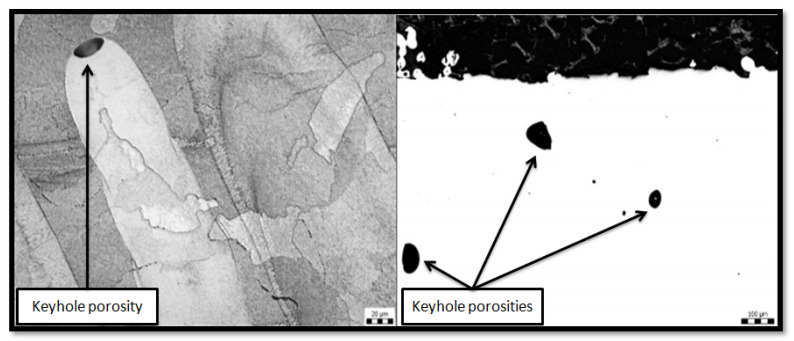
Keyhole porosity at the end of laser track.

**Figure 9 materials-14-06527-f009:**
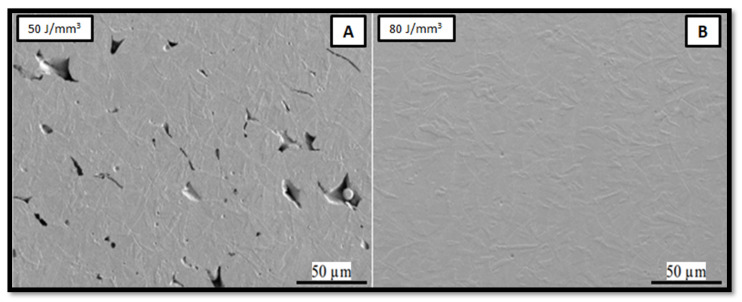
SEM images of porosity at (**A**) 50 and (**B**) 80 J/mm^3^ [25].

**Figure 10 materials-14-06527-f010:**
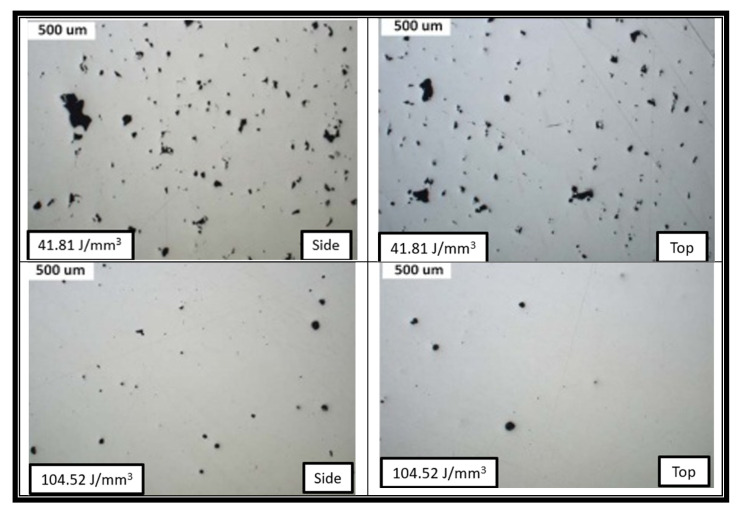
Porosity at different energy densities of 41.81, 104.52, and 209.03 J/mm^3^.

**Figure 11 materials-14-06527-f011:**
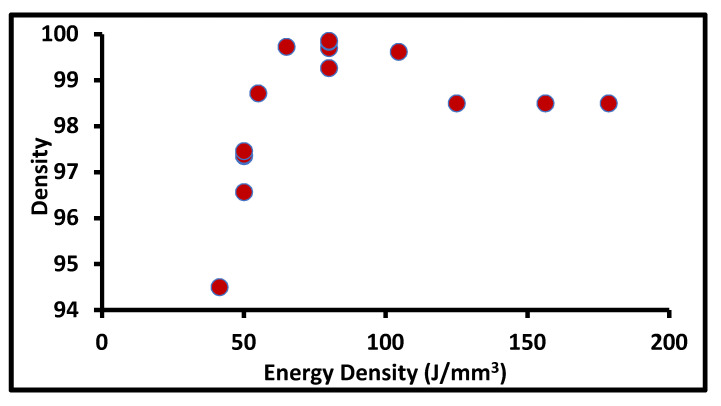
The density of the build part at a different energy density.

**Figure 12 materials-14-06527-f012:**
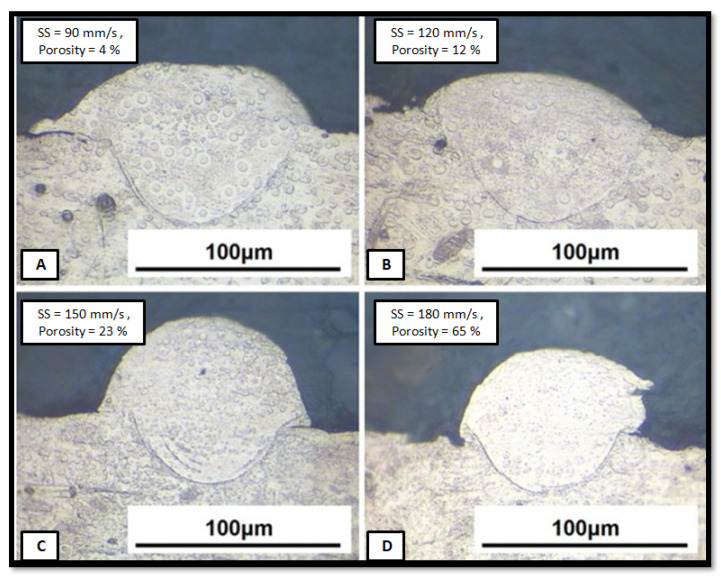
Molten tracks at different scanning speed: (**A**) 90 mm/s (**B**) 120 mm/s (**C**) 150 mm/s (**D**) 180 mm/s with corresponding porosities of 4%, 12%, 23%, and 65% [9].

**Figure 13 materials-14-06527-f013:**
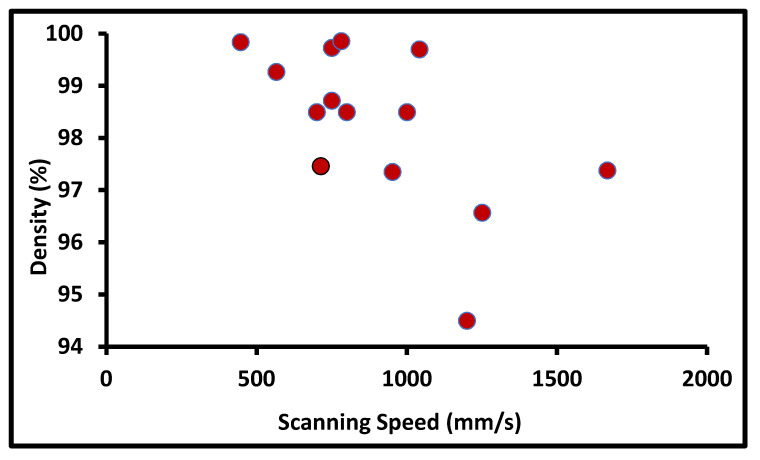
Density at a different scanning speed.

**Figure 14 materials-14-06527-f014:**
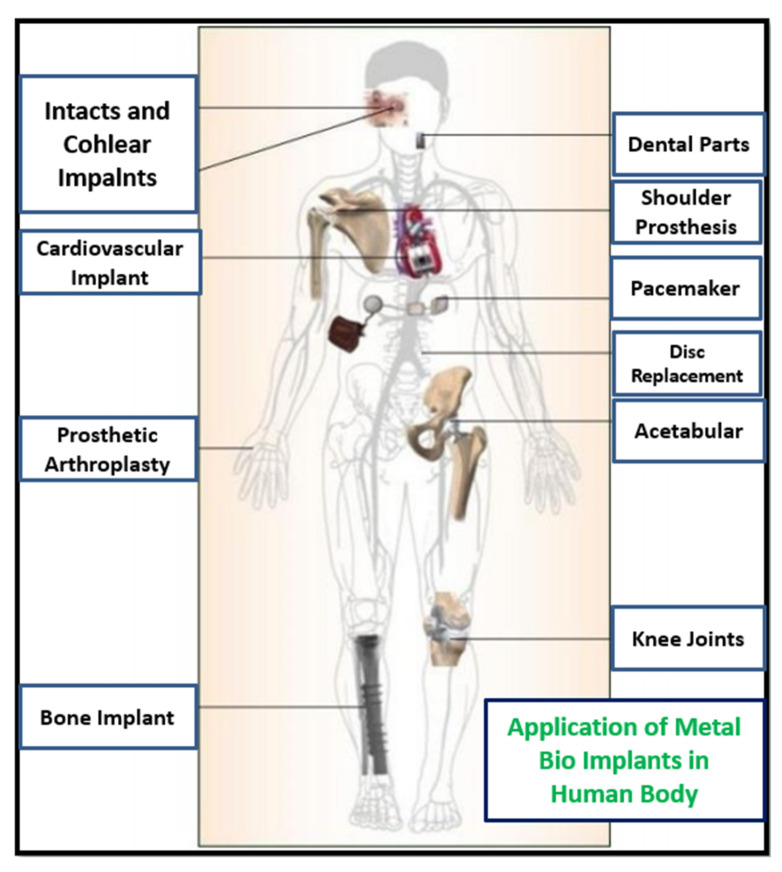
Application of metal bio implant in different areas of the human body [42].

**Figure 15 materials-14-06527-f015:**
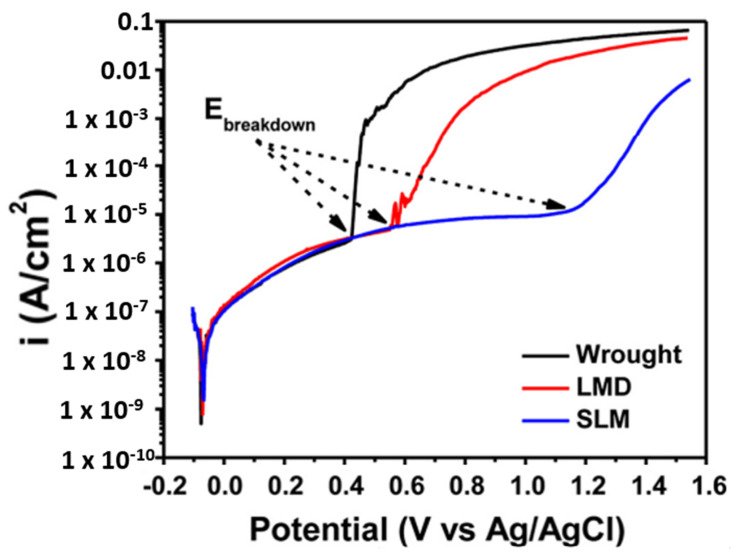
Corrosion measurement by potentiodynamic polarization curve of wrought, LMD, and SLM SS316L [37].

**Figure 16 materials-14-06527-f016:**
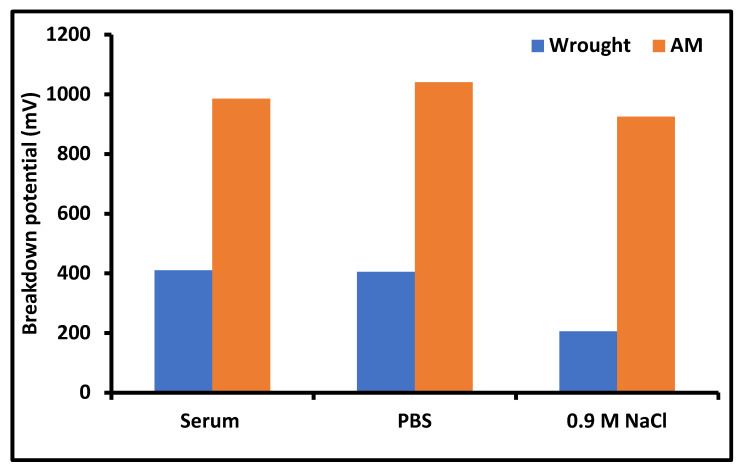
Breakdown potential in different environments of wrought and AM parts [54].

**Figure 17 materials-14-06527-f017:**
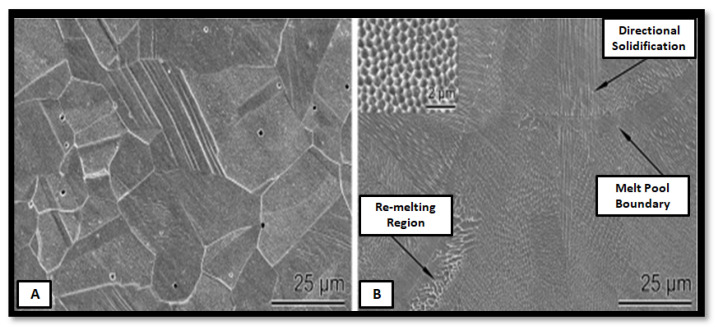
Microstructure of SS316L: (**A**) wrought and (**B**) AM parts [40].

**Figure 18 materials-14-06527-f018:**
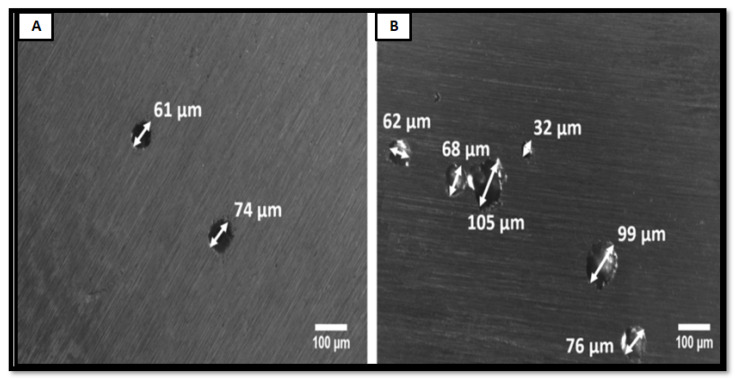
Pit formation in corrosion testing (**A**) SLM and (**B**) wrought SS316L [42].

**Figure 19 materials-14-06527-f019:**
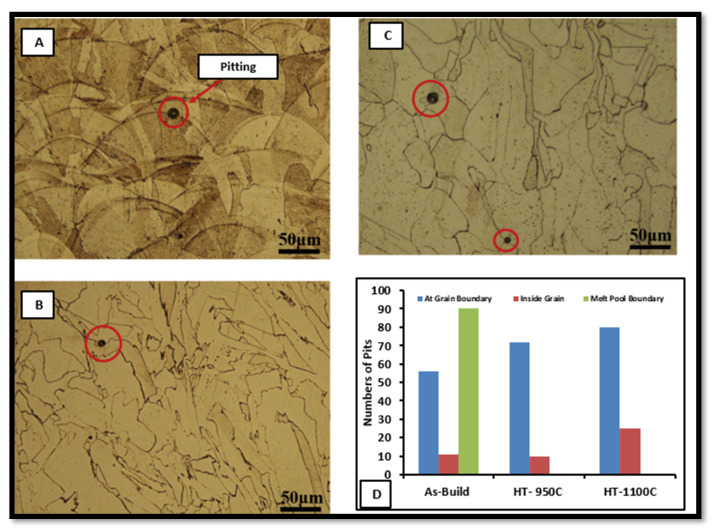
Micrographs and pit formation at (**A**) as-built, (**B**) HT 950’C, (**C**) HT 1100. (**D**) Number of pits at different conditions [45].

**Figure 20 materials-14-06527-f020:**
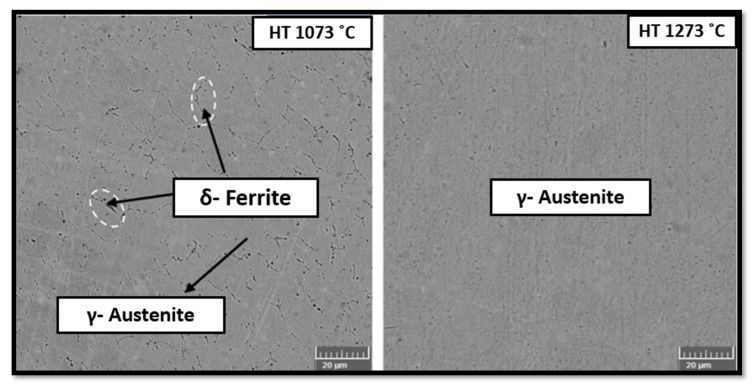
SEM images at different HT temperature.

**Figure 21 materials-14-06527-f021:**
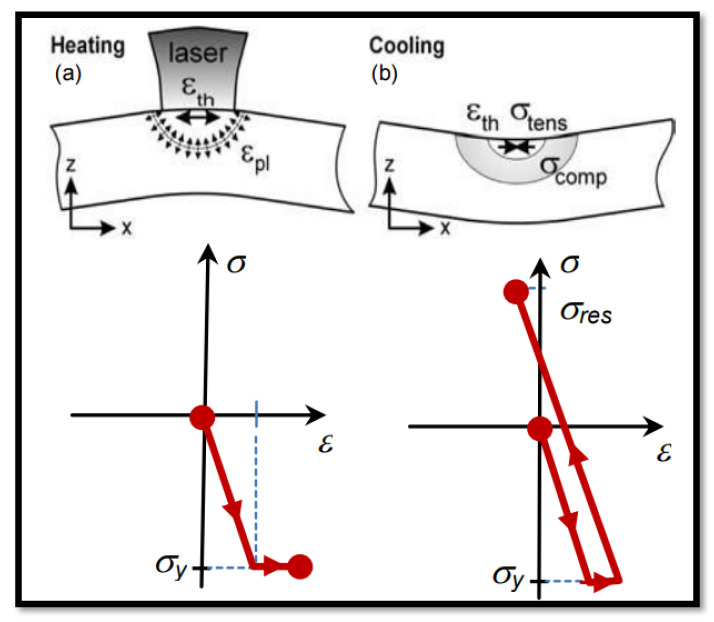
Residual stress generation (**a**) Heating, (**b**) Cooling [72].

**Figure 22 materials-14-06527-f022:**
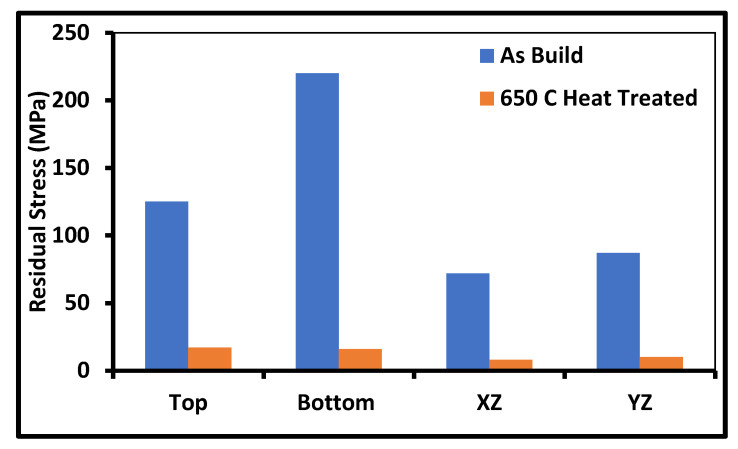
Residual stress: as-built vs. post-heat treatment of SS316L [35].

**Table 1 materials-14-06527-t001:** Tensile strength at different process parameters.

No.	Laser Power (w)	Layer Thickness (µm)	Hatch Spacing (µm)	Scanning Speed (m/s)	Energy (J/mm^3^)	UTS (MPa)	Reference
1	200	20	100	1	100	594	[15]
2	300	30	80	0.7–1.2	-	590	
3	90	30	150	1	20	621.7	
4	200	50	110	0.75	48.48	684.2	
5	380	50	120–360	0.187–0.25	-	550–700	
6	100	50–100	80	0.1–0.3	-	500–600	
7	100,200	50	-	0.20–0.22	-	662–750	
8	200	30	60	2	55.55	611.9	[10]
9	200	30	60	2		589	
10	200	30	60	2		597.6	
11	300	30	80	0.7	178.57	530	[11]
12	300	30	80	0.8	156.25	580	
13	300	30	80	1	125	575	
14	300	30	80	1.2	104.17	570	
15	200	50	110	0.65	55.94	614	[10]
16	200	50	110	1.2	30.3	171	
17	107	30	200	0.4	44.5	580	[16]
18	107	30	200	0.5	35.67	446	
19	107	30	200	0.5	35.67	405	
20	107	30	200	0.4	44.58	572	
21	107	60	200	0.4	22.3	231	
22	107	60	200	0.4	22.3	242	
23	107	60	200	0.5	17.83	115	
24	107	60	200	0.5	17.83	147	
25	230	30	-	0.8	-	720	[10]
26	175	30	120	0.75	64.8	UTS With Angle Rotation in range of 640 Mpa	[17]
27	100	30	90	0.55	67.3		
28	200	30	120	0.8	69.4		
29	100	30	90	0.4	92.6		

**Table 2 materials-14-06527-t002:** Effect of different process parameters on hardness.

No.	Laser Power (W)	Layer Thickness (µm)	Hatch Spacing (µm)	Scanning Speed (m/s)	Powder Size (Micron)	Energy (J/mm^3^)	Hardness (HV)	Reference
1	180	50	124	-	-	125.42	225 HV	[26]
2	300	30	80	0.7	-	178.57	267	[11]
3	300	30	80	0.8	-	156.25	272.5	
4	300	30	80	1	-	125	278	
5	300	30	80	1.2	-	104.17	258	
6	107	30	200	0.4	-	44.58	104 HRB	[17]
7	107	30	200	0.4	-	44.58	92	
8	107	60	200	0.4	-	22.29	76	
9	107	60	200	0.4	-	22.29	71	
10	107	30	200	0.5	-	35.66	78	
11	107	30	200	0.5	-	35.66	86	
12	107	60	200	0.5	-	17.83	48	
13	107	60	200	0.5	-	17.83	45	
14	200	30	60	2	16 µm	55.55	XY 276	[27]
15	200	30	60	2			XZ 291	
16	200	30	60	2			YZ 286	
17	200	30	60	2	4–48 µm		XY 281	
18	200	30	60	2			XZ 246	
19	200	30	60	2			YZ 249	
20	200	30	60	2	48 µm		XY 277	
21	200	30	60	2			XZ 248	
22	200	30	60	2			YZ 255	
23	200	50	-	1.6	15–40 µm	-	XY 262	[15]
24	200	50	-	1.6		-	XZ 237	
25	200	50	-	1.6		-	YZ 239	
26	380	50	25–120	3	20–63 µm	-	213–220	
27	180	50	124	0.557–1.670	15–45 µm	-	235	
28	100–150	20	50–70	0.7	15–45 µm	-	210–240	
29	100–200	50	-	0.2–0.22	20–63 µm	-	247–255	
30	300	30	80	1	-	125	281.6	[11,26,28]
31	300	30	80	1.2	-	104.17	260	
32	180	50	124	0.231	-	125	225	
33	380	50	120	0.65	-	101.33	213	
34	380	50	25	3	-	101.33	220	
35	190	20–30	40	0.8	-	198–297	325	
36	200	50	100	1	-	200	239	
37	90	25	80	187	-	58.4	240	
38	200	30	80	1.042	-	80	213	[26]
39	150	30	80	1.25	-	50	185	
40	380	50	25	3	-	101.33	Hardness decreases with hatch spacing increases Maximum hardness at 25 µm hatch spacing, i.e., 220	[28]
41	380	50	30	2.5	-	101.33		
42	380	50	35	2	-	108.57		
43	380	50	40	1.75	-	108.57		
44	380	50	50	1.5	-	101.33		
45	380	50	60	1.25	-	101.33		
46	380	50	70	1.05	-	103.4		
47	380	50	80	0.95	-	100		
48	380	50	90	0.85	-	99.35		
49	380	50	100	0.75	-	101.33		
50	380	50	110	0.7	-	98.7		
51	380	50	120	0.625	-	101.33		

**Table 3 materials-14-06527-t003:** Effect of different process parameters on relative density (as-built).

No.	Laser Power (w)	Layer Thickness (µm)	Hatch Spacing (µm)	Scanning Speed m/s	Energy Density (J/mm^3^)	Relative Density (%)	References
1	300	30	80	700	178.57	98.5	[11]
2	300	30	80	800	156.25	98.5	
3	300	30	80	1000	125	98.5	
4	300	30	80	1200	104.16	94.5	
5	350	50	110	650	97.90	99.6	[16]
6	150	30	80	1250	50	96.57	[25]
7	200	30	80	1667	49.99	97.38	
8	150	30	140	714	50.02	97.46	
9	200	30	140	952	50.02	97.35	
10	150	30	120	750	55.55	98.72	
11	150	30	80	1133	55.16	98.59	
12	175	30	120	750	64.81	99.73	
13	200	30	140	1198	39.74	99.24	
14	150	30	80	781	80	99.86	
15	200	30	80	1042	79.97	99.7	
16	150	30	140	446	80.07	99.84	
17	200	30	140	565	84.28	99.27	
18	90	25	56	1500	42.85	95	[35]
19	90	25	84	600	71.42	99.1	
20	90	25	84	300	142.85	99.25	
21	90	25	84	300	142.85	99	
22	100	60	100	90	185.18	96	

## Data Availability

Not applicable.

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
