# Peer review of "A Critical Review on Effect of Process Parameters on Mechanical and Microstructural Properties of Powder-Bed Fusion Additive Manufacturing of SS316L"

_materials, 2021, doi:10.3390/ma14216527_

Round 1

Reviewer 1 Report

The authors made a detailed review on powder-based AM of SS316L. However, this manuscript needs to be revised significantly before publication. The comments are mentioned below:

  1. The title mentions 'powder based AM'. I think it should be 'powder bed fusion' since it is more written on LPBF studied. There is very little discussion on powder based DED.
  2. Line 64: 'Bio-implant'-'b' should be small.
  3. Figure 1: A1, A2, B1, B2 caption details should be mentioned in the figure title.
  4. Line 120: Why the author kept upper limit of optimum energy density as 105J/mm^3? UTS seems to constant upto 150 J/mm^3. If there is more keyhole formation then how the UTS is similar above 105 J/mm^3?
  5. "The SLM process generates a denser structure compared to the direct energy deposition (DED) process"-I don't agree with this comment. DED of SS316L is very common and based on optimized process parameter you can get similar or even denser parts fabricated using DED.
  6. Figure 5,6,7 are not discussed by calling those figure numbers on text. They should discuss mentioning these figures so that readers can corelate the results.
  7. Figure 10: The scale bar in the figures are difficult to read. Please update them.

Author Response

Dear Reviewer,

we would like to thank you for taking your time to read and comment on our manuscript. In the attachment, you can find our response. 

Reviewer 2 Report

  • Line 14: Re-write keeping the mind the formal style requirements of academic writing. 
  • Line 17: Need to specify the AM process used, at the beginning itself. 
  • Line 19: Check international AM standards (ISO/ASTM) for nomenclature. The process name is 'laser powder bed fusion.  Also, cite the standards in the introduction chapter while defining the process. 
  • Line 20: Residual stress is not a process parameter, it is the effect. Are the authors focussed on mechanical properties/microstructure or residual stresses? 
  • Line 21: Not sure what it means in the context. 
  • Line 23: electrochemical? I am unable to comprehend the context
  • Line 24: SS 316L or SS316L? Choose one, preferably the latter
  • Line 26: Space missing
  • Line 27: Relative density or porosity? It is a completely different topic/effect to study for LPBF, same as residual stress. The focus of the work is not clearly defined/presented
  • Line 28/29: Again off-topic
  • Line 30: Residual stress generation? What does it mean?
  • Line 31: different Am techniques? or LPBF? Off-topic again
  • Line 32/33: I am not sure what human implants are and how do they matter in this review. Kindly explain
  • Line 34/35: which 'quality? I thought we are focused on mechanical properties and microstructure. 
  • Line 26: Re-write keywords. SLS does not produce SS 316L. I see 'corrosion behaviour' is added as a keyword, then why is that not discussed properly in the abstract. Which corrosion properties are the authors considering? No discussion of bio-implants in the abstract, which implants? which process? which properties? 
  • Line 39: Demands of AM? Re-write

I read the title of the paper and abstract, and I am not able to understand the aim and objectives of this review. The title, abstract, and keywords need a major overhaul, and so does the entire article. I am outlining few major issues that I found below: 

  • Line 62-64: Authors mention keywords used for literature study. Is this a systematic review or a focussed state-of-the-art review? Kindly decide and write accordingly. 
  • The introduction section ends very poorly. Does not define clearly what the next sections talk about, what the outcome of the review will be? How will it help the AM community? 
  • Figure 1: I am not sure if the citation, reference and the figure itself correspond with each other. 
  • Figure 2: Is fracture behaviour even the focus of this review?  
  • 21. and 2.2 are the same? Why are the section titles the same? 
  • Missing reference: Figure 3, 5, 6, 7, , 10, 11, 13, 14
  • A complete table on hardness (Table 2). why? is this the objective of the review? 
  • table 3. Which density? 
  • Chapter 3: which implants? are SS 316L implants even used in the human body? where are the references? I think most of the implants are of titanium alloys. Kindly check the literature. 

Overall: 

  • Kindly spend some time reading state-of-the-art literature review articles on AM and laser powder bed fusion to understand few things such as nomenclature, the difference between process parameters and the effect of those parameters, different application areas, materials used in those application areas. 
  • Serious work needs to be done in improving the writing style, sentence formation, logical paragraph formation, discussions, and conclusions. 

Author Response

(The authors gave the same response as above.)

Round 2

Reviewer 1 Report

The authors made necessary changes to the manuscript. 

Author Response

Dear Reviewer, 

Thank you very much for your positive opinion. 

Yours Sincerely, 

Authors

Reviewer 2 Report

Authors still have left a lot of comments unaddressed and have not worked on the holistic message that this critical review is supposed to give. 

I am sorry but I have to reject the manuscript in its current state. 

Author Response

Dear Reviewer,

We have already addressed all our comments, explanation, and corrections in our last revision. Despite that, you did not give any specific comments which we should improve.  

Below we put all our comments which we attached in the 1st round review: 

Comment 1: Line 14: Re-write keeping the mind the formal style requirements of academic writing.
Response: Thank you for your valuable inputs. In response to the reviewer’s comment, the sentence has been rewritten in the revised manuscript. (Please see Page No. 1). Additive manufacturing (AM) is one of the recently focused research areas due to its ability to eliminate different subtractive manufacturing limitations, such as difficultly in fabricating complex parts, material wastage, and numbers of sequential operations.

Comment 2:Line 17: Need to specify the AM process used, at the beginning itself.
Response: In the revised manuscript the details pertaining to AM process have been mentioned at the beginning, as follows:
“The laser-powder bed fusion (L-PBF) AM for SS 316L known for complex part production due to layer-by-layer deposition, hence it is extensively used in the aerospace, automobile, and medical sectors”.

Comment 3:Line 19: Check international AM standards (ISO/ASTM) for nomenclature. The process name is 'laser powder bed fusion. Also, cite the standards in the introduction chapter while defining the process.
Response: Thank you for the clarification. As per the comment, we have re-checked the ASTM standards to understand basic terminology related to all AM processes. Thus, the manuscript has been updated with all the basic terminology focused on the Powder bed fusion process. In addition to this, the ASTM standards have been cited at the beginning of the introduction while
defining the process as follows: “Powder bed fusion AM uses thermal energy to selectively fuse powder particles on powder bed as mentioned in ASTM standard [3].” [3] ISO/ASTM, Additive Manufacturing - General Principles Terminology (ASTM52900), Rapid Manuf. Assoc. (2013) 10–12. https://doi.org/10.1520/F2792-12A.2.

Comment 4:Line 20: Residual stress is not a process parameter, it is the effect. Are the authors focussed on mechanical properties/microstructure or residual stresses? Response: Thank you for your valuable inputs. In response to the comment. We have modified the entire abstract to clearly define our objective and focused area. In this article, we have primarily focused on the effect of process parameters on mechanical and microstructure properties. However, as the residual stress is also one of the important factors in LPBF-AM, we
have also presented the effect of residual stresses on mechanical properties as well in the manuscript. The abstract has been revised as follows in the updated manuscript: 
This review critically elaborates the effect of various input parameters, i.e., laser power, scanning speed, hatch spacing, and layer thickness on various mechanical properties of AM SS316L such as tensile strength, hardness, and relative density along with microstructure evolution. The effect of other AM parameters such as build orientation, pre-heating temperature, particles size on build properties was also discussed. The scope of this review is also concerned with the challenges in practical application. Hence, the residual stress formation, their influences on mechanical properties, and corrosion behavior of AM build a part for bio implant application is also discussed. This review involves a detailed comparison of properties achievable with different AM techniques and various post-processing techniques such as heat treatment and grain refinement effects on properties. This review would help in selecting
suitable process parameters for various human body implants and many different applications. This study would also help to better understand the effect of each process parameter of PBFAM on SS316L builds part quality.

Comment 5:Line 21: Not sure what it means in the context.
Response: Thank you for the comment. We have rewritten the abstract with clear the scope and objective of the review study. The part of the updated abstract is as follows: This review critically elaborates the effect of various input parameters, i.e., laser power, scanning speed, hatch spacing, and layer thickness on various mechanical properties of AM SS316L such as tensile strength, hardness, and relative density along with microstructure evolution. The effect of other AM parameters such as build orientation, pre-heating temperature, particles size on build properties was also discussed. The scope of this review is also concerned with the challenges in practical application. Hence, the residual stress formation, their influences on mechanical properties, and corrosion behavior of AM build a part for bio implant application is also discussed. This review involves a detailed comparison of properties achievable with different AM techniques and various post-processing techniques such as heat
treatment and grain refinement effects on properties. This review would help in selecting suitable process parameters for various human body implants and many different applications. This study would also help to better understand the effect of each process parameter of PBFAM on SS316L builds part quality.

Comment 6:Line 23: electrochemical? I am unable to comprehend the context
Response: The electrochemical studies mean, the corrosion behavior of LPBF-AM build sample here i.e. Corrosion current density, break down potentials. However, as per suggestion, the change has been made in the manuscript as follows: “The scope of this review is also concern the challenges in practical application. Hence, the residual stress formation, their influences on mechanical properties and corrosion behavior of AM build a part for bio implant application is also discussed.”

Comment 7:Line 24: SS 316L or SS316L? Choose one, preferably the latter
Response: It is SS316L, we have re-checked the entire manuscript and updated it with the correct one “SS316L”. Thank you for the clarification.

Comment 8:Line 26: Space missing
Response: Thank you. We have checked spacing-related mistakes in the entire paper and corrected them accordingly.

Comment 9:Line 27: Relative density or porosity? It is a completely different topic/effect to study for LPBF, same as residual stress. The focus of the work is not clearly defined/presented
Response: The main focus of this review is to study how the porosity in AM manufactured build parts influences mechanical properties. We have re-defined the focus of this review paper in the updated abstract also have re-written the sentence to avoid confusion between relative density and porosity as follows,
This review critically elaborates the effect of various input parameters, i.e., laser power, scanning speed, hatch spacing, and layer thickness on various mechanical properties of AM SS316L such as tensile strength, hardness, and effect of porosity along with microstructure evolution. Also, the porosity in AM build structure plays an important role in deciding the overall mechanical properties of the building part. Hence the effect of porosity is also considered under the scope of this review to relate the mechanical properties. The detailed discussion for the importance of porosity is mentioned in section 2.3 as follows,
The porosity of the built part is one of the important properties to be considered, as directly or indirectly it influences all the mechanical and microstructure behavior. In additive manufacturing, densification of material is the most crucial factor to be considered. The optimum range of process parameters is required to get the denser product. In this section, the
formation of porous, affecting process parameters and their influences optimizes parameters in terms of mechanical and microstructural properties.

Comment 10:Line 28/29: Again off-topic
Response: Thank you for the clarification, we have made changes in the abstract with improved objective and scope of this review in the revised manuscript. (Please see Page No. 1)

Comment 11:Line 30: Residual stress generation? What does it mean?
Response: Residual stress generation, here we referred to the evolution of residual stresses within build materials during layer by layer deposition. We have covered how residual stresses are formed during LPBF-AM in the second paragraph of section 4. We have also revised the sentence in the updated manuscript as follows. The scope of this review is also concerning the challenges in practical application. Hence, the residual stress formation, their influences on mechanical properties, and corrosion behavior of AM build a part for bio implant application is discussed.

Comment 12: Line 31: different Am techniques? or LPBF? Off-topic again
Response: This article focused on the LPBF of SS316L. In this article, we have also attempted to compare the different mechanical properties for different additive manufacturing techniques such as Powder bed fusion and direct energy deposition for better understanding.

Comment 13:Line 32/33: I am not sure what human implants are and how do they matter in this review. Kindly explain

Response: 1) Human Implant is one of the major applications of SS316L as its biocompatible material.

2) Corrosion property for implant application is the major challenge as reported in Chew et al. [43].
3) In the AM, as the microstructure evolution is totally different from the conventional
manufacturing processes, the corrosion behavior also differs as mentioned in [52].
Hence in this article, we are correlating these three points for a discussion of how the
additively manufactured sample involves different corrosion behavior. How the LPBF
improves the corrosive properties compared to conventional manufacturing processes. The
given references are used for consideration for the application of SS316L for bio implants.
43. K. Chew, S. Hussein, S. Zein, and A. L. Ahmad, “The corrosion scenario in the human body :
Stainless steel 316L orthopaedic implants,” vol. 4, no. 3, pp. 184–188, 2012, doi:
10.4236/ns.2012.43027.
44. A. Pandey, A. Awasthi, and K. K. Saxena, “Metallic implants with properties and latest
production techniques : a review,” Adv. Mater. Process. Technol., vol. 00, no. 00, pp. 1–36,
2020, doi: 10.1080/2374068X.2020.1731236.
45. G. Manivasagam, D. Dhinasekaran, and A. Rajamanickam, “Biomedical Implants :
Corrosion and its Prevention -A Review Biomedical Implants : Corrosion and its Prevention -
A Review,” no. June, 2010, doi: 10.2174/1877610801002010040.
49. M. Sivakumar and S. Rajeswari, “Investigation of failures in stainless steel orthopaedic
implant devices : pit-induced stress corrosion cracking,” vol. 11, pp. 1039–1042, 1992.
52. L. Bai, C. Gong, X. Chen, Y. Sun, J. Zhang, and L. Cai, “Additive Manufacturing of
Customized Metallic Orthopedic Implants :Materials , Structures , and Surface Modifications,”
pp. 1–25, 2019, doi: 10.3390/met9091004.

Comment 14:Line 34/35: which 'quality? I thought we are focused on mechanical properties and microstructure
Response: Yes, in this article we are focusing on mechanical properties and microstructure. This article helps to know the effect of process and would help to achieve desired properties. Hence, the quality refers to the overall betterment of the build part properties.

Comment 15:Line 26: Re-write keywords. SLS does not produce SS 316L. I see 'corrosion behavior' is added as a keyword, then why is that not discussed properly in the abstract. Which corrosion properties are the authors considering? No discussion of bio-implants in the abstract,
which implants? which process? which properties?
Response: Thank you for your valuable inputs, we have implemented and updated the keywords as follows:,
Powder bed fusion, Process parameters, corrosion behavior, residual stresses, bio implant This review focused on the different aspects of additive manufacturing pertaining to practical applications of SS316L. Here, the bio implant is one of the major applications of SS316L (as mentioned in Response 13 also). Hence, we have also covered the corrosion behavior of
additively manufactured as compared to conventional manufacturing. The entire Section 3 is discussed about the different bio implant studies reported for AM part of SS316L, their corrosion properties, bio implant failure, and their concerns, how the PBF-AM have a positive impact on bio implant applications and the research gaps need to focus in future studies. The entire abstract is also updated as mentioned in Response 5 which clearly defines the scope and
focused area of this article.

Comment 16:Line 39: Demands of AM? Re-write
Response: Thank you for pointing out, we have rewritten the sentence in the updated manuscript as follows:
The applications of additive manufacturing are increasing exponentially, especially in the medical[1] and aerospace due to its unique feature of fabricating complex geometrical components. Powder industry[2], bed fusion AM uses thermal energy to selectively fuse powder particles on powder beds as mentioned in the ASTM standard [3]. Comment 17:Line 62-64: Authors mention keywords used for literature study. Is this a systematic review or a focused state-of-the-art review? Kindly decide and write accordingly.

Response: Thank you for continuously helping in improving our article. Yes, this is a state-of-the-art review. We have removed the keywords statement in the updated manuscript, which was mentioned just for reference. 

Comment 19:The introduction section ends very poorly. Does not define clearly what the next sections talk about, what the outcome of the review will be? How will it help the AM community?
Response: The last paragraph of the introduction sections explains the flow of the literature review, the different process parameters, and mechanical properties, and also how this review helps AM community in defining the process parameters for desired properties. The details are
specifically contained in the last paragraph of the introduction with the following partial text: At the beginning of the section, the effect of process parameters on tensile strength, hardness, and porosity with microstructure evolution is explained. and then, the corrosion and residual stress
behavior of additively manufactured SS316L with process parameters are described. All the reported process parameters on AM-SS316L and their influences are considered to conclude. 

This study would help to define the process parameter window for future applications to get desired strength and hardness. This study also helps to collect all the process parameters effect reported till now for AM-SS316L to future work of machine learning for optimization of process parameters.

Comment 20:Figure 1: I am not sure if the citation, reference, and the figure itself correspond with each other.
Response: Figure 1explain the effect of powder particle size on the tensile strength, as
mentioned. However, the statement is updated as follows,
Chen et al.[9] examined the effect of powder particles on mechanical properties with an energy density of 55.55 J/mm3 and observed that finer particles around ~16 µm give the highest tensile strength in the range of 610 MPa as shown in Fig.1. The fracture surface with dimples as shown in Fig.1 A2 and B2 also implies the ductile failure of the sample.

Comment 21:Figure 2: Is fracture behavior even the focus of this review?
Response: The energy density affects the fracture behavior of the built sample which influences the tensile strength. Hence, it has been discussed in the manuscript

Comment 22:21. and 2.2 are the same? Why are the section titles the same?
Response: Thank you for the clarification. We have revised the manuscript and updated it with the correct terms.
2.1 Tensile strength
2.2 Hardness
Comment 23: Missing reference: Figure 3, 5, 6, 7, , 10, 11, 13, 14
Response: We have collected the process parameters and particular output properties from the reported literature, to understand the range of process parameters and their influence on particular properties. Hence, we have developed Fig: 3,5,6,7,10,11,13 based on the collected data. The set of data are also mentioned in Table 1, Table 2, and Table 3.
The citation for the Figure 14 is updated as follows.
45. Ali, S.; Rani, A.M.A.; Baig, Z.; Ahmed, S.W.; Hussain, G.; Subramaniam, K.; Hastuty,
S.; Rao, T.V.V.L.N. Biocompatibility and corrosion resistance of metallic biomaterials.
Corros. Rev.2020, 38, 381–402, doi:doi:10.1515/corrrev-2020-0001.

Comment 24:A complete table on hardness (Table 2). why? is this the objective of the review?
Response: The hardness value reported for the LPBF-AM of SS316L has been collected from the reported literature. This data has been plotted into the different graphs with respect to input parameters such as hardness vs energy density, hardness vs layer thickness, hardness vs hatch spacing as mentioned in Figure 5,6,7. The data collection helped to find out the minimum
and maximum hardness achieved for LPBF-AM and also to understand the effect of each parameter on hardness.

Comment 25:table 3. Which density?
Response: The density refers to the relative density of build part. The caption has been revised in the updated manuscripts as follows;
Table 1. Effect of different process parameters on relative density (as build)

Comment 26:Chapter 3: which implants? are SS 316L implants even used in the human body? where are the references? I think most of the implants are of titanium alloys. Kindly check the literature.

Response:Yes, Titanium is also used as bio implant application for the human body. However, SS316L has been extensively used from many years for different human implant applications such as, hip joints, knee joints, screws and plates. The reference articles have been also cited in revised manuscript as follows;
44. Chew, K.; Hussein, S.; Zein, S.; Ahmad, A.L. The corrosion scenario in human body :
Stainless steel 316L orthopaedic implants. 2012, 4, 184–188,
doi:10.4236/ns.2012.43027.
45. Ali, S.; Rani, A.M.A.; Baig, Z.; Ahmed, S.W.; Hussain, G.; Subramaniam, K.; Hastuty,
S.; Rao, T.V.V.L.N. Biocompatibility and corrosion resistance of metallic biomaterials.
Corros. Rev.2020, 38, 381–402, doi:doi:10.1515/corrrev-2020-0001.
46. Pandey, A.; Awasthi, A.; Saxena, K.K. Metallic implants with properties and latest
production techniques : a review. Adv. Mater. Process. Technol.2020, 00, 1–36,
doi:10.1080/2374068X.2020.1731236.
47. Manivasagam, G.; Dhinasekaran, D.; Rajamanickam, A. Biomedical Implants :
Corrosion and its Prevention -A Review Biomedical Implants : Corrosion and its
Prevention - A Review. 2010, doi:10.2174/1877610801002010040.
49. Mccafferty, E. Effect of Ion Implantation on the Corrosion Behavior of Iron , Stainless
Steels , and Aluminum — A Review. 2001, 57, 1011–1029.
50. Okazaki, Y.; Gotoh, E. Comparison of metal release from various metallic biomaterials
in vitro. 2005, 26, 11–21, doi:10.1016/j.biomaterials.2004.02.005.
51. Sivakumar, M.; Rajeswari, S. Investigation of failures in stainless steel orthopaedic
implant devices : pit-induced stress corrosion cracking. 1992, 11, 1039–1042.
52. Hemmasian Ettefagh, A.; Guo, S.; Raush, J. Corrosion performance of additively
manufactured stainless steel parts: A review. Addit. Manuf.2021, 37, 101689,
doi:10.1016/j.addma.2020.101689.
53. Eliaz, N. Corrosion of metallic biomaterials: A review. Materials (Basel). 2019, 12.
54. Bai, L.; Gong, C.; Chen, X.; Sun, Y.; Zhang, J.; Cai, L. Additive Manufacturing of
Customized Metallic Orthopedic Implants : Materials , Structures , and Surface
Modifications. 2019, 1–25, doi:10.3390/met9091004.

As you can see we provide our comments regarding your review, we would be grateful for pointing us which parts have not been sufficiently improved. 

Yours sincerely, 

Authors

Round 3

Reviewer 2 Report

I have read the revisions. Thanks for that. I apologize but I will stick with mu decision. Unfortunately the manuscript does not add any significant knowledge to the journal or the research community.